

# Rossby wave packets driving concurrent and non-concurrent heatwaves in the Northern and Southern Hemisphere mid-latitudes

Maria Pyrina[1,2], Wolfgang Wicker[2], Andries Jan de Vries[2], Georgios Fragkoulidis[3,4], and Daniela I.V. Domeisen[2,1]

[1]Institute for Atmospheric and Climate Science, ETH Zurich, Zurich, Switzerland
[2]Institute of Earth Surface Dynamics, University of Lausanne, Lausanne, Switzerland
[3]Institute for Atmospheric Physics, Johannes Gutenberg University Mainz, Mainz, Germany
[4]Institute for Environmental Research and Sustainable Development, National Observatory of Athens, Athens, Greece

**Correspondence:** Maria Pyrina (maria.pyrina@env.ethz.ch)

**Abstract.** Heatwaves that occur simultaneously over several regions, termed concurrent heatwaves, pose compounding threats to society and the environment. Amplified quasi-stationary circumglobal Rossby wave patterns (CGWPs) and high-amplitude transient non-circumglobal Rossby Wave Packets (RWPs) have been proposed as two possible explanations for the occurrence of heatwaves. The relation of these mechanisms for heatwaves has been investigated over different timescales, but their rel-

evance for concurrent and non-concurrent heatwaves remains unexplored. In the present study we focus on daily time scales and investigate the relevance of the global CGWP amplitude and of the local RWP amplitude for the occurrence of concurrent and non-concurrent heatwaves over the Northern Hemisphere (NH) and Southern Hemisphere (SH) mid-latitudes. To distinguish between concurrent and non-concurrent heatwaves we apply a k-means clustering algorithm on all heatwaves detected in ERA5 reanalysis data within the 1959–2021 period. We identify 42 spatial clusters of heatwaves in the NH and 53 in the

SH. In all identified clusters, mid-latitude heatwaves typically occur at the leading edge of RWPs where Rossby wave breaking takes place in the form of ridge building or block formation. No specific zonal wavenumber is more frequently related to the concurrent or to the non-concurrent heatwave category. However, for high global CGWP amplitudes concurrent heatwaves occur more often in the NH when the dominant zonal wavenumber is k = 7, and non-concurrent heatwaves occur more often in the SH for k = 5. The mid-latitude regions exhibiting increased heatwave probabilities under the influence of either global or

local high wave amplitude, include western North America, central Europe, Black Sea, Tibet, the southwest coast of Australia, as well as the southern Indian and Atlantic Oceans. Over those regions, the local high amplitude RWPs increase heatwave probabilities by a factor ranging from 4 to 7, whereas the maximum factor for high global CGWP amplitude is 2. These results emphasize the importance of the daily RWP amplitude and the weak association of the global CGWP amplitude to heatwave occurrence over the NH and SH mid-latitudes. This research for the first time investigates the underlying atmospheric dy-

namical processes that contribute to the development of concurrent and non-concurrent heat extremes, a crucial step towards improving our understanding and ability to predict heatwave variability at weather and longer time scales.





## 1 Introduction

Heatwaves can have disastrous impacts on societies worldwide, affecting human health, agriculture, and infrastructure (Shaposhnikov et al., 2014; van der Velde et al., 2010; McEvoy et al., 2012). Heatwave impacts can be exacerbated during concurrent heatwave events, that is, when heatwaves occur simultaneously over several regions worldwide (Vogel et al., 2019). Specifically, impacts can become particularly severe when heatwaves occur simultaneously in regions with high exposure of people or crops (Vogel et al., 2019; Kornhuber et al., 2020). The examination of atmospheric mechanisms leading to concurrent heatwaves versus non-concurrent heatwaves can, therefore, provide important insights into potential differences with respect to their physical drivers and predictability.

The occurrence of concurrent heatwaves could be driven by one atmospheric flow feature affecting all heatwave regions, e.g., circumglobal Rossby wave patterns (CGWPs), or by mechanisms that can act at the same time without necessarily exhibiting a clear interrelation. Examples of such local mechanisms leading to heat extremes encompass blocking highs and Rossby wave packets (RWPs), both of which do not necessitate the presence of circumglobal waves and have been linked to the occurrence of non-concurrent heatwaves. As RWPs can trigger extreme surface weather over multiple synoptic wavelengths, they may be responsible for the manifestation of extremes in multiple locations.

RWPs consist of synoptic-scale troughs and ridges confined to a limited longitudinal range and exhibit a wave amplitude that decays away from its center in longitude (Wirth et al., 2018; Chang, 2001). The zonally varying amplitude of RWPs is referred to as the RWP envelope, which propagates eastward with the RWP group velocity. The eddies embedded within the RWP propagate with their individual phase velocities, which are typically smaller than the group velocity (Chang, 1993; Esler and Haynes, 1999). Therefore, the envelope of RWPs moves eastward faster than individual troughs and ridges, reflecting the dispersive nature and downstream development of RWPs (Rossby, 1945; Wirth et al., 2018). RWPs have been linked to high impact weather, and specifically to heatwaves (Wirth et al., 2018; Davies, 2015; Fragkoulidis et al., 2018). RWPs that amplify intermittently at the same longitudes for a prolonged period of time are termed *recurrent Rossby wave packets* and studies have shown that their occurrence can lead to extreme events of increased persistence (Röthlisberger et al., 2019; Ali et al., 2022).

The occurrence of hot extremes collocated with blocking highs has also been well documented in observations (Barriopedro et al., 2011; Pfahl and Wernli, 2012; Black et al., 2004). Blocks are persistent anticyclonic atmospheric flow patterns that can last for several days or even weeks (Drouard et al., 2021). One of the processes conducive to blocking formation is Rossby wave breaking (RWB) (Pelly and Hoskins, 2003; Altenhoff et al., 2008; Masato et al., 2012; Pfahl et al., 2015). RWB is usually associated with the final stage of the RWP life cycle, with the wave energy being dispersed in the meridional direction favoring the onset of blocking highs (Wirth et al., 2018).

RWPs and blocking highs often evolve symbiotically and interact with each other. In particular, RWPs can reinforce blocking highs or ridges downstream, by anchoring the location of the block or ridge and sustaining its life. The reinforcement takes place via transient eddy vorticity forcing upstream of blocking anticyclones (Berggren et al., 1949; Shutts, 1983) or initiation upstream of a pre-existing ridge (Röthlisberger et al., 2018). Recurrent RWPs can also foster blocking downstream of the amplifying waves and blocking highs can act as a metronome for the mid-latitude Rossby waves further downstream leading to





recurrent RWPs (Röthlisberger et al., 2019). The latter is due to the stationarity of blocking leading to recurrent RWP conditions downstream, as troughs are expected to form repeatedly at similar longitudes downstream of a block.

An alternative mechanism suggested to be associated with the incidence of non-concurrent and, to a greater extent, concurrent extremes is quasi-resonant wave amplification (QRA), which involves the existence of CGWPs. Specifically, it is proposed that when certain dynamical conditions on the nature of the mid-latitude waveguide are fulfilled, synoptic-scale waves can resonate with circumglobal quasi-stationary waves and thereby induce hemispheric, persistent, and large-amplitude circumglobal Rossby wave patterns (Petoukhov et al., 2013; Coumou et al., 2014; Kornhuber et al., 2017a). Note that in the literature CG-WPs are also termed *circumglobal wave patterns, circumglobal Rossby waves, circumglobal waves, circumglobal wave trains,* and *circumglobal teleconnection patterns.*

Studies exploring the connection between QRA and heatwaves reveal that an increase in the amplitude of particular zonal wavenumbers of quasi-stationary CGWPs increases the frequency of concurrent heatwaves. Regarding the NH, heatwaves over North America, Western Europe and the Caspian Sea region were related to amplified wave events of zonal wavenumbers k=5–8 due to QRA (Teng et al., 2013; Coumou et al., 2014; Kornhuber et al., 2019, 2017a). Regarding the SH, QRA of quasi-stationary Rossby waves was found important for HW occurrence over Patagonia and Australia, and related to the amplification of the zonal wavenumbers k=4 and k=5 (Kornhuber et al., 2017b). However, studies exploring the influence of transient Rossby waves, specifically RWPs, to heatwaves have also connected the zonal wavenumbers k=4 and k=5 to days of high RWP activity leading to heatwaves in southeast Australia (Ali et al., 2022).

Our investigation explores the significance of both the global and local daily wave amplitude in the context of concurrent and non-concurrent heat extremes. Using a k-means clustering algorithm, we categorize heatwaves into these two groups, with a specific focus on the mid-latitudes of the NH and SH. In Section 2, we present the data and methods used in this study. In Section 3, we present the results and discussion. Specifically, in subsection 3.1, we present the structure of RWPs, RWB, and zonal wind during concurrent and non-concurrent heatwaves. Subsequently, in subsection 3.2 we explore how the amplitude of specific zonal wavenumbers in the global wave amplitude is associated with increased probabilities of concurrent and non-concurrent heat extremes in the two hemispheres. In subsection 3.3 we identify the regions where the probabilities of concurrent and non-concurrent heatwaves increase during high global and local RWP amplitudes.

## 2 Data and Methods

### 2.1 Data

For all the analyses in this study we use the ERA5 reanalysis data for the period 1959–2021 (Hersbach et al., 2020). The 2-meter temperature (t2m) data were obtained at a 3-hourly resolution on a $0.5° \times 0.5°$ latitude-longitude grid. The 300hPa meridional wind (v300) and zonal wind (u300) data were obtained at a 6-hourly resolution on a coarser $2° \times 2°$ latitude-longitude grid, as the dynamics investigated in this study are not expected to change for a finer spatial grid resolution. For each variable we obtain the daily mean. Daily anomalies are calculated with respect to the 29-day running mean of the daily climatology for the reference period 1961-1990. The global surface temperature anomalies show temperatures to be relatively stable from





1950 until the late 1970s and drastically increase afterwards (Bell et al., 2021). In order to take into account the non-linear
global warming trend over the period 1959–2021 and remove spurious trends that may exist due to the drastic increase in the
number of assimilated observations in ERA5 after 1979 (Hersbach et al., 2020), we linearly detrended the anomaly fields at
the grid scale separately for the periods 1959-1979 and 1980-2021 (Fig. A1). In the following, the results are reported based
on detrended variables. The analysis was repeated without detrending and also with detrending using different methods and
consistent results were found.

## 2.2    Heatwave days and k-means clustering algorithm

Heatwaves are investigated during the respective summer months of the two hemispheres, i.e., June, July, August for the NH,
and December, January, February for the SH. Heatwave days are identified per grid point and are defined as days when the daily
t2m detrended anomaly exceeds the 95th percentile of that day's climatological distribution. Similar to Russo and Domeisen
(2023) and Perkins-Kirkpatrick and Lewis (2020), the daily t2m detrended anomaly distribution is constructed by considering
all the days of the period 1959–2021 falling in a 29-day window centered on the day of interest. The sensitivity of our analysis
to heatwave persistence is investigated with the application of the clustering algorithm (see section 2.3).

For each grid point in the study region we assign a value of 1 to the heatwave (HW) days and a value of 0 to the non-
heatwave days. The k-means clustering algorithm is then applied on the resulting temporally-varying binary field (Pedregosa
et al., 2011), separately for the NH (30°N–70°N) and SH mid-latitudes (30°S–70°S). The grid points are provided to the
algorithm as a vector, while the time dimension represents the number of samples of the grid point vector. Therefore, applying
the algorithm in the time dimension means that each time step (sample) is assigned to a cluster. K-means clustering is a
non-hierarchical clustering method and, contrary to hierarchical clustering methods, reallocates points (days) that may have
been mis-grouped at an earlier stage. The algorithm starts by computing the centroids for each cluster and then calculates the
distances between each data vector and each of the centroids, assigning the vector to the cluster whose centroid is closest to it.
This process is repeated until all data vectors are assigned to a cluster containing members that are closer to its centroid than
to the centroids of other clusters. A disadvantage of the k-means algorithm is the need to pre-specify the number of clusters
(k), which may compromise its ability to reassign potentially mis-classified data points (Wilks, 2011). Unless there is prior
knowledge of the correct number of clusters, the k-means clustering should be repeated for a range of cluster numbers in order
to assess its robustness.

We evaluate the algorithm's output across a range of cluster numbers (C=10-70). For each number of clusters, the algorithm
generates C-1 clusters with a nearly equal distribution of data and one dominant *majority* cluster, encompassing a high percent-
age of days. The majority cluster either represents frequently occurring and specific HW patterns or a set of days that cannot
be assigned to a specific cluster and form a distinct cluster of their own. If the majority cluster signifies a frequently occurring
pattern (i.e., a high number of heatwave days over the locations indicated by the cluster centroid), increasing the cluster number
should result in similar cluster centroids for the new clusters. However, our findings indicate that increasing the cluster number
leads to clusters with substantial centroid differences. Choosing a high cluster number (e.g., C=70) leads to a high number of



clusters with a low number of days assigned (i.e., less than 1 percent), whereas choosing a low cluster number (e.g., C=10) leads to the majority cluster having too many days.

The optimal cluster number we set is the one that corresponds to the lowest percentage of assigned days to the majority
cluster and at the same time to the highest number of clusters with more than 1 percent of days assigned to each of them (Appendix Fig. A2). Clusters with less than 1 percent of days assigned to them have a low frequency of HW day occurrence and, therefore, are not investigated in this study. For the NH the optimal cluster number we find is C=42, and for the SH C=53.

HWs with large spatial extent will be more likely tied to distinct atmospheric circulation patterns and prominent impacts. Moreover, as we are interested in synoptic-scale heatwaves, the minimum HW spatial extent we set for HW regions is 320 grid
points, which for our resolution of a $0.5° \times 0.5°$ latitude-longitude grid is equal to at least 354,524 km$^2$. Our minimum HW spatial extent is roughly double than the 151,000 km$^2$ used by Lyon et al. (2019) investigating heatwaves over the United States of America. If a cluster covers more than one region with at least 320 HW grid points, then we categorise it as a concurrent heatwave cluster, otherwise it is defined as a non-concurrent heatwave cluster.

For the NH we find 16 concurrent clusters and 12 non-concurrent clusters with more than 1 percent of heatwave days as-
signed. The heatwave clusters are named after the main heatwave occurrence region, being the region with the largest heatwave spatial extent. Therefore, in the NH we find 5 North American clusters (NAm), 2 North Atlantic clusters (NAt), 7 European clusters (EU), 7 Asian clusters (Asia), and 7 Pacific clusters (Pac). In the SH we obtain 19 concurrent clusters and 7 non-concurrent clusters, comprising of 10 Pacific clusters (Pac), 3 South American clusters (SAm), 3 South Atlantic clusters (SAt), 1 African cluster (AF), 4 Indian Ocean clusters (IOc), 2 Australian clusters (AU), 1 New Zealand cluster (NZ), and 2 Southern
Ocean clusters (SOc). Due to the large number of clusters, in the following we focus on clusters with the most frequent heatwave occurrence regions over land. We show the results for the most frequent clusters per region, except for the case of the NH concurrent, for which we show clusters with a prominent global signal of RWPs.

Repeating the same analysis for the 35°N–65°N latitudinal belt in the NH and the 35°S–65°S latitudinal belt in the SH leads to similar preferred cluster numbers in the NH and SH (N=40, N=50). Moreover, the concurrent and non-concurrent clusters
found are similar to the ones found for the 30°–70° regions, due to the heatwave locations occurring mainly at the center of the study regions and not at the edges of the selected latitude range.

## 2.3 Clustering of persistent heatwaves

Heatwaves are usually defined as events with at least three consecutive heatwave days (Perkins, 2015; Perkins-Kirkpatrick and Lewis, 2020; Russo and Domeisen, 2023). To explore whether persistence has an effect on our cluster analysis we applied
k-means clustering after filtering for a minimum heatwave persistence of three heatwave days. In the NH, the optimum cluster number is reached for C=74, which results into 33 percent of days assigned to the majority cluster and to 29 clusters representing more than 1 percent of heatwave days each. From these 29 clusters, 19 are classified as concurrent clusters and 10 as non-concurrent clusters. Overall, the clusters we find for a 3-day persistence criterion are similar to those for a non-persistence requirement in the NH, but include a smaller percentage of heatwave days. A high number of clusters is needed in order to
reach the optimum cluster number (C=114) in the SH as well.





Clustering with a heatwave persistence criterion of at least 3 heatwave days leads to a sample size reduction of 49 percent for the NH and 55 percent for the SH. To identify meaningful clusters and generalize, clustering algorithms, including k-means, rely on the assumption that the given dataset captures the key characteristics and trends in a larger dataset. Therefore, a reduced sample size can affect the convergence of the algorithm (Craen et al., 2006). This might be the reason why for C=74 in the
NH and C=114 in the SH, the algorithm still assigns 33 and 61 percent of days into the majority clusters of the NH and SH, respectively. To achieve a better representation of the regions that have predominantly experienced heatwaves during the summers of 1959-2021, we investigate in the following the clusters of heatwaves without the persistence criterion. To account for the persistent nature of heatwaves, we also provide information on the persistence of the heatwaves included in each of the identified clusters.

**2.4 Global and local Rossby wave amplitude**

The horizontal scale of Rossby waves varies seamlessly between planetary (k=1–3) and synoptic scale (k>3) wavenumbers, with the range of zonal wavenumbers k=4–9 including the synoptic-scale waves commonly involved in circumglobal wave amplification during heatwaves (Petoukhov et al., 2013; Coumou et al., 2014; Kornhuber et al., 2017a). We calculate the amplitude of the daily global CGWP amplitude, by summing the amplitudes of wavenumbers k=4–15 for each hemisphere.
To determine the amplitude of each zonal wavenumber, first we apply a decomposition of the v300 field. The Fast Fourier Transform algorithm (FFT) is applied for every latitude at every given time t in the longitude dimension. The amplitude $A_k$ is given by the area-weighted latitudinal mean of the absolute value of the discrete Fourier transform (DFT) coefficients for the respective zonal wavenumber k and normalized by the number of longitude points. More specifically, for the real function $v(x,t)$ (Eq. 1), being the meridional wind velocity with $0 < x \leq 2\pi$:

$$v(x,t) = A(t)cos(kx + \Phi(t)) \tag{1}$$

the DFT (Eq. 2)) is computed as follows at every given t:

$$\hat{v_k} = \sum_{l=1}^{N} v(x)e^{-ikl2\pi/N} \tag{2}$$

where A(t) > 0 the amplitude of the wave, k = 1, 2, 3, ... the non-dimensional zonal wavenumber, $\Phi(t)$ is the phase of the wave in radians at time t and l = 1,...,N the longitudinal grid point index for the number of longitudes N.
A measure of the local Rossby wave amplitude is the Rossby wave packet envelope $RWP_{env}$, which we calculate following the method by Fragkoulidis et al. (2018). This method diagnoses RWPs after implementing a number of refinements on the Hilbert transform method of Zimin et al. (2003) and uses the semi-geostrophic coordinate space following the approach of Wolf and Wirth (2015), resulting in a reduction of spurious fragmentation of RWPs. For each day the $RWP_{env}$ is calculated as a daily mean over the $RWP_{env}$ values computed at 6-hourly time steps using v300 detrended anomalies and detrended values
of geopotential height at the 300hPa level.





## 2.5 Detection of Rossby wave breaking

Common indicators for RWB are potential vorticity (PV) streamers and cutoffs (Wernli and Sprenger, 2007). PV streamers refer to elongated structures of the dynamical tropopause at quasi-horizontal isentropic surfaces, while isolated high or low PV air masses are commonly known as cutoffs (Hoskins et al., 1985). Studies that examined the relation between RWB and

extreme surface weather have mostly focused on extreme precipitation and used stratospheric PV streamers and cutoffs as indicators of RWB (Massacand et al., 1998; Moore et al., 2019; De Vries, 2021). Stratospheric PV air masses exhibit PV > 2 PVU in the NH and PV < -2 PVU in the SH (1 potential vorticity unit; 1 PVU = $10^{-6} K \ kg^{-1} \ m^2 \ s^{-1}$). In this study, we link the occurrence of heatwaves to RWB, and therefore use tropospheric PV structures ($PV_{trop}$) that can also be considered as a proxy for atmospheric ridges and blocks (Nakamura, 1994; Schwierz et al., 2004; Rohrer et al., 2018). Tropospheric PV

structures are characterized by low PV air masses, that is PV < 2 PVU in the NH and PV > -2 PVU in the SH.

For the identification of PV streamers and cutoffs, we follow the method introduced by Wernli and Sprenger (2007) and some of the adaptations from De Vries et al. (2018) and De Vries (2021). Next, we briefly summarize the algorithm used for identifying tropospheric PV streamers and cutoffs, including a description of the additional criteria for the PV cutoffs identification. PV is computed on model levels and then interpolated onto isentropic surfaces. The ERA5 data used for the PV

computations were derived on a 0.5°×0.5° latitude-longitude grid at 6-hourly time steps.

First, we define the dynamical tropopause by the ±2 PVU contours at 18 isentropic surfaces between 275K and 360K with 5K intervals. Next, we determine the so-called stratospheric PV reservoir by the most equatorward +2 PVU contour in the NH (-2 PVU in the SH) that encircles the globe. Tropospheric PV streamers are identified by elongated structures with low PV air masses that satisfy the following geometric criteria: (i) length > 1,000 km, (ii) width < 1,500 km, (iii) ratio of length over

width > 1, and (iv) the length along the contour between the start and end points of the potential PV streamer < 15,000 km (the latter after Sprenger et al. (2013)). Furthermore, we remove all tropospheric PV streamers at isentropic surfaces if more than 50 percent of the stratospheric reservoir is classified as stratospheric PV streamers to avoid spurious features at isentropic surfaces with heavily distorted ±2 PVU contours.

Tropospheric PV cutoffs are identified based on closed ±2 PVU contours, that is, tropospheric PV air masses embedded

within stratospheric air at isentropic surfaces. To filter out tropospheric PV cutoffs that are associated with radiative and frictional processes near the surface rather than dynamical processes (i.e., RWB), we remove (i) any tropospheric PV cutoff that intersects the surface topography, and (ii) any tropospheric PV cutoff without having the stratospheric body aloft (i.e., on the isentropic surface above), while allowing for vertically connecting (i.e., spatially overlapping) PV cutoffs. Consistent with the latter criterion, tropospheric PV cutoffs are only identified on isentropic surfaces until 355K. Also, small-scale tropospheric

PV cutoffs with a surface area below $25 \times 10^3 \, \text{km}^2$ are removed to focus on synoptic-scale structures only.

For each day, we compute the daily mean vertical depth of tropospheric PV structures ($PV_{trop}$) from the number of isentropic surfaces on which PV structures are detected. Each PV structure represents a vertical extent of 5K, consistent with the vertical spacing of the isentropic surfaces.





## 3  Results and Discussion

### 3.1  Large-scale circulation during concurrent and non-concurrent heatwaves

In the following, we analyze the role of Rossby wave packets, Rossby wave breaking, and zonal wind for concurrent and non-concurrent heatwave occurrence. The results are presented separately for the Northern and Southern Hemisphere mid-latitudes.

#### 3.1.1  Northern Hemisphere

The composites of standardized temperature anomalies for the summers of 1959–2021 are shown for the 8 selected concurrent and non-concurrent heatwave clusters of the NH in Figure 1. These clusters correspond to the concurrent clusters: EU2, PAC3, NAt2, NAm2 and to the non-concurrent clusters: EU7, PAC7, Asia5, NAm4 shown in Figure A3. In general, heatwaves occur more frequently in concurrent clusters, with 40 percent of heatwave days occurring in concurrent clusters versus 35 percent occurring in non-concurrent clusters (Fig. A3). The remaining 25 percent of heatwave days are attributed either to clusters with very low heatwave occurrence frequency (<1 percent), or are not attributed at all to a specific cluster (see methods). The regions of heatwave occurrence are indicated by the contour level +0.3 (solid black line). For the non-concurrent clusters we observe one region of heatwave occurrence (right column Fig. 1), whereas for the concurrent clusters (left column Fig. 1) we observe at least 2 regions in the NH with standardized temperature anomalies above +0.3 standard deviations, denoting regions with frequent simultaneous heatwaves for the particular clusters.

To demonstrate the atmospheric processes associated with concurrent clusters (Fig. 2) and non-concurrent clusters (Fig. 3), we construct composites of the meridional (v300) and zonal wind (u300) anomalies, as well as composites of the RWP amplitude ($RWP_{env}$). Heatwave locations exhibit negative anomalies in u300 (right hand column in Fig. 2, Fig. 3), suggestive of the regions on the southern flank of an anticyclone or ridge (Bluestein, 1992). Rossby waves are horizontally non-divergent, so they are apparent not only in the meridional, but also in the zonal wind. Therefore, for most of the European and North American concurrent and non-concurrent clusters there are intensified westerlies north and south of the heatwave location (Fig. A4, Fig. A5). This zonal wind structure is sometimes referred to in literature as a double jet. This characteristic is not visible in the mean zonal upper-level circulation (see Figure 6.4 in Hartmann (2015)) and it was previously linked to several extra-tropical extreme weather events (Kornhuber et al., 2017b; Rousi et al., 2022; Xu et al., 2021). Contrary to the studies stating that the double jet is responsible for extreme weather, other studies support that the Rossby waves themselves contribute to the double jet feature (Wirth and Polster, 2021). Therefore, the double jet may be a consequence of, rather than a precondition for, large wave amplitudes (Wirth and Polster, 2021).

Over the heatwave location of all concurrent and non-concurrent clusters, a pair of high amplitude southerly and northerly winds (positive/ negative values) emerges in the respective composite of v300, indicating the presence of a ridge over the heatwave location. Sequences of southerly and northerly winds in the v300 composites reflect the presence of so-called wave trains. The v300 wave trains cover larger regions of the NH in the case of the concurrent clusters, compared to the non-concurrent clusters, with either single or multiple RWPs being related to the occurrence of concurrent heatwaves. For example, a single RWP is responsible for the concurrent heatwaves that occur simultaneously over North America and Europe in the



clusters EU and NAt, whereas different RWPs are responsible for the heatwaves found in the Pac cluster over the Pacific, North America, North Atlantic, and Northeast Asia.

The main heatwave location in each heatwave cluster is found in the vicinity of regions with enhanced Rossby wave breaking (RWB) (Fig. 4). The co-location of RWB and heatwaves indicates that there is strong meridional overturning of the circulation at that location, consistent with the presence of a ridge or block. Warming can be expected by the poleward advection of warm air at the upstream flank of $PV_{trop}$ structures and by subsidence (adiabatic warming) collocated with the $PV_{trop}$ structures. Regarding Asian heatwaves, different types of wave breaking (i.e., cyclonic warm, cyclonic cold, anticyclonic warm, anticyclonic cold; see also Masato et al. (2012)) were associated with the formation of summer blocking highs over several Asian regions (Shi et al., 2016).

In general, the locations of v300 wave trains, RWPs, and RWB in concurrent and non-concurrent clusters do not change for 1-day and 3-day heatwave persistence thresholds, even though the sample size decreases to almost half (Figures A6, A7). The clusters that demonstrate persistent to non-persistent heatwave day ratios higher than 50 percent include all concurrent clusters, except for the cluster NAm2, and more than half of the non-concurrent clusters, except for the North American clusters and the Asia6 cluster (Figure A8). While the EU1 cluster is not as frequent as the European clusters EU3 and EU7, it stands out as the most persistent cluster, with heatwaves persisting more than 3 days for approximately 79 percent of its days.

### 3.1.2 Southern Hemisphere

Similar to the NH, in the SH, the concurrent clusters demonstrate a higher total frequency of occurrence than the non-concurrent clusters, specifically, 39 percent versus 20 percent (Appendix, Fig. A10). The duration of extremes is different in the two hemispheres. The SH clusters exhibit lower persistence compared to the clusters in the NH, with most of the clusters demonstrating persistent to non-persistent heatwave day ratios lower than 50 percent (Appendix Fig. A8). The higher persistence of extremes in the NH compared to the SH is related to the different characteristics of synoptic scale circulation, such as higher blocking frequency and lower RWP phase speed in the NH (Toulabi Nejad et al., 2022; Fragkoulidis and Wirth, 2020). Moreover, the role of topography in NH is important, as topography located along the mid-latitude jet increases heatwave frequency upstream of the topographic forcing (Jiménez-Esteve and Domeisen, 2022). Finally, thermodynamic effects like the drying of land in the NH can also exacerbate heatwaves in the NH mid-latitudes (Seo and Ha, 2022; Sutanto et al., 2020; Miralles et al., 2019).

The standardized temperature anomalies of the selected concurrent and non-concurrent heatwave clusters of the SH are shown in Figure 5. These clusters correspond to the concurrent clusters: Pac3, AU, IOc2, SAm1 and to the non-concurrent clusters: SAt1, NZ, AU, AF shown in Figure A10. As in the NH, most of the SH heatwaves, e.g, AU heatwaves in Fig. 5, occur at the leading edge of RWPs, in-between bands of southerlies and northerlies (Fig. 6), and mainly over areas with high RWB frequency (Fig. 8). This spatial configuration of RWP-RWB is observed for most of the land and marine heatwaves shown in Figure 5. Previous studies have linked anticyclonic RWB to heatwaves in southeastern Australia, occurring due to warm air advection from the interior of the continent to the coasts (Engel et al., 2013). Moreover, recurrent RWPs have been associated with a significant increase in persistence of hot spells over land areas of the SH, such as southern Australia and South America (Ali et al., 2022).





## 3.2 Dominant zonal wavenumbers during concurrent and non-concurrent heatwaves

Several studies showed that the intensification of the amplitude of particular zonal wavenumbers of quasi-stationary CGWPs
increases the heatwave occurrence probabilities over some regions of the two hemispheres. We investigate the connection
between heatwave probabilities and the dominant zonal wavenumbers identified through FFT in the daily CGWP amplitude
(global wave amplitude). The dominant zonal wavenumber is defined as the zonal wavenumber that exhibits the highest am-
plitude on a specific day and, therefore, has the highest contribution to the global wave amplitude. We provide the climato-
logical distribution of the dominant zonal wavenumbers for the daily v300 anomaly values during the summers 1959–2021
(purple bar), as well as the climatological distribution of the dominant wavenumbers during concurrent (black line) and non-
concurrent (black dot) heatwave clusters (Fig. 9). Additionally, the dominant wavenumbers for concurrent (orange line) and
non-concurrent (orange dot) heatwave clusters are provided with respect to days with global wave amplitude anomalies higher
than +1.5 standard deviations.

The two most prevalent daily summer wavenumbers climatologically are the wavenumbers k=5 and k=6 in both hemispheres
and for both concurrent and non-concurrent heatwaves. Moreover, the climatological probability of occurrence of the dominant
zonal wavenumbers is similar to their estimated probability for either concurrent or non-concurrent heatwave clusters. Thus,
there is no specific wavenumber occurring more frequently in neither of the two heatwave cluster categories. This result
is confirmed by a two-sided student's t-test, applied separately for each zonal wavenumber to calculate whether there is a
statistically significant difference in its mean occurrence at the 0.05 significance level. Specifically, we use a bootstrapping
method and sub-sample from the climatological distribution a number of samples equal to the number of days of concurrent
or non-concurrent clusters (e.g., 2308 for the concurrent heatwaves of the NH) and we repeat that for 100 times. The same
method was applied to calculate whether there is a statistically significant difference between the probability densities of the
concurrent (non-concurrent) heatwave clusters with and without the occurrence of high wave amplitude.

For high global wave amplitude days there is a statistically significant increase of only the wavenumber k=7 for concurrent
heatwave clusters in the NH and of only the wavenumber k=5 for non-concurrent heatwave clusters in the SH. One of the
concurrent clusters of the NH where the zonal wavenumber k=7 appears most frequently is the NAm cluster. Specifically, we
find an occurrence frequency of k=7 equal to 23 percent versus 20 percent for climatology (Fig. 9). The concurrent heatwaves
of the NAm cluster occur over North America, Western Europe and the Caspian Sea region, in similar regions as the ones
reported in Kornhuber et al. (2019) to occur frequently with amplified k=7 events (see their Figure 2). In the SH, high heatwave
occurrence in Patagonia (Fig. A11) and Australia (Fig. 6) is observed in clusters with a high occurrence frequency of k=5 (39
percent), surpassing the climatological occurrence rate of 35 percent.





### 3.3 Relevance of global and local high wave amplitude for heatwave occurrence

In the previous section we explored the connection of heatwaves to high amplitude wave events, as such events are related to
heatwaves with prominent impacts (Kornhuber et al., 2020). We saw that when CGWPs attain high amplitudes, they project more frequently onto the zonal wavenumber k=7 for concurrent heatwaves in the NH and k=5 for non-concurrent heatwaves in the SH. In this section we identify the regions where heatwave probabilities increase during high global and local wave amplitudes. We aim to determine whether the two metrics, global and local, lead to qualitatively similar spatial patterns of concurrent and non-concurrent heatwave probabilities.

The conditional probability of heatwave occurrence with respect to high global wave amplitude days (daily CGWP amplitude anomaly higher that +1.5 standard deviations) is estimated for every grid point using the Equation 2.10 of Wilks (2011). Specifically, the estimated conditional heatwave probability is equal to the number of heatwave days co-occurring with high wave amplitude days, divided by the sum of high amplitude days. The unconditional heatwave probability is calculated via the count of all heatwave days over all days. Regarding the estimation of conditional probabilities for concurrent (non-concurrent)
heatwaves, we take into account at every grid point only the concurrent (non-concurrent) heatwave days and the high global wave amplitude days for concurrent (non-concurrent) clusters. Regarding the estimation of unconditional concurrent (non-concurrent) heatwave probabilities, we consider the count of concurrent heatwave days over all days considered in our study. The same methodology is followed to calculate the conditional probabilities of heatwaves during high local wave amplitude. A high local wave amplitude day occurs when the grid point $RWP_{env}$ value surpasses +1.5 $RWP_{env}$ standard deviations and
occurs together with a ridge over that location.

The ratio between the conditional heatwave probabilities for high global and local wave amplitude to the unconditional heatwave probabilities in the NH mid-latitudes is provided for all heatwaves as well as for the concurrent and non-concurrent heatwaves in Figure 10. Probability ratios with values larger (smaller) than 1 indicate that heatwaves are more (less) likely to occur in the presence of large wave amplitude. Heatwave probabilities are higher during high local wave amplitude than during
high global wave amplitude days (Fig. 10 panels (d), (a), respectively). Specifically, heatwave probabilities increase by up to a factor of 2 when the global amplitude metric is used, versus by up to a factor of 7 for the local wave amplitude metric. The highest heatwave probabilities are observed over west North America, Greenland, and Tibet (Fig. 10d).

High amplitude local RWPs increase the occurrence probabilities of concurrent and non-concurrent heatwaves by more than a factor of 10 over several mid-latitude regions of the NH. The maximum probability ratios for concurrent heatwaves are
observed over the areas of Alaska, the northwest Pacific coast, Greenland, Tibet, and northeast Asia. The maximum probability ratios for non-concurrent heatwaves occur over Canada, west United States, Greenland, central Europe, north of the Caspian Sea, Tibet, and central-west Asia. Over central Europe and the Balkan region, the presence of high amplitude RWPs leads to 10 times and 13 times higher occurrence probabilities of concurrent and non-concurrent heatwaves, respectively.

The fact that global and local high wave amplitude signals show different probability values and patterns over the mid-
latitudes, points to the fact that heatwaves are strongly modulated by RWPs. In the SH mid-latitudes we can draw the same conclusion for the relationship of global versus local wave amplitude to heatwaves (Fig. 11). In the SH, the maximum prob-



ability ratios related to high amplitude local RWPs are found over the southeast coast of Australia, as well as the southern Indian and Atlantic Oceans. Over those regions the heatwave probabilities in relation to the local RWP metric increase by approximately a factor of 14 for concurrent and 20 for non-concurrent heatwaves.


## 4    Conclusions

The main objective of this study is to identify the contribution of both the global and the local Rossby wave amplitude for the occurrence of concurrent and non-concurrent heatwaves over the mid-latitudes of the Northern Hemisphere (NH) and Southern
Hemisphere (SH).

First, we cluster the mid-latitude heatwaves of the NH and SH into the regions where they most frequently occur. We separate the heatwaves into concurrent and non-concurrent according to their temporal alignment with heatwaves among different locations. In the NH, 40 percent of heatwave days occur in concurrent heatwave clusters and 35 percent in non-concurrent heatwave clusters. The other 25 percent is either not clustered or falls into clusters we do not investigate due to their low
heatwave frequency. In the SH, concurrent heatwaves occur at a similar frequency as in the NH, being 39 percent, whereas SH non-concurrent heatwaves have an occurrence frequency of only 20 percent.

Both concurrent and non-concurrent heatwaves in the mid-latitudes develop mainly at leading edge of Rossby wave packets (RWPs), in between bands of southerlies and northerlies, collocated with increased frequency of Rossby wave breaking. Composites of meridional wind anomalies show that wave trains formed during heatwaves cover larger regions of the NH and SH
mid-latitudes in the case of the concurrent clusters compared to the non-concurrent clusters. In some cases, such as for one of the Pacific concurrent heatwave clusters of the NH, it is not the spatial extent of RWPs that leads to concurrent heatwaves, but the fact that RWPs appear simultaneously over different regions.

In a second step we explore the global wave amplitude signal and its projection on specific zonal wavenumbers climatologically and also in relation to heatwaves. We show that the dominant zonal wavenumber occurrence probability follows the
climatological summer distribution in both hemispheres and for both concurrent or non-concurrent heatwave clusters. This result indicates that there is no single zonal wavenumber more related to one or the other heatwave category. However, for high global wave amplitude days we find a statistically significant increase of the zonal wavenumber k=7 in the NH during concurrent heatwave clusters and of k=5 for non-concurrent heatwave clusters in the SH.

Our findings regarding the dominant zonal wavenumbers during high amplitude global wave events and their relation to
increased heatwave probabilities over several mid-latitude regions, agree with studies that use 15-day and 30-day averages of temperature and meridional wind. These studies suggest that the stationarity of almost circumglobal waves is a dominant factor for the co-occurrence of heat extremes. We demonstrate that the occurrence of concurrent heatwaves is connected to RWPs of large spatial extend and to RWPs that occur simultaneously over different regions. Furthermore, despite certain mid-latitude regions exhibiting high heatwave probabilities being common to both global and local high wave amplitude, the overall





heatwave probability values and patterns related to the two metrics differ. This result emphasizes the weak association of the global wave amplitude to heatwave probabilities over the NH and SH mid-latitudes.

High amplitude local RWPs increase the occurrence probabilities of concurrent heatwaves by a factor of 15 over the regions of Alaska, the northwest Pacific coast, Greenland, Tibet, and northeast Asia. Non-concurrent heatwave probabilities increase by a factor of 18 in the presence of high amplitude local RWPs over Canada, west United States, Greenland, north of the Caspian

Sea, and Tibet. Over the SH, the maximum increase in concurrent and non-concurrent heatwave probabilities related to high amplitude local RWPs is found over the southeast coast of Australia, as well as the southern Indian and Atlantic Oceans.

Overall, we find that RWPs are driving the occurrence of concurrent and non-concurrent heatwaves in the mid-latitudes of the two hemispheres. Therefore, a detailed understanding of the relevant dynamical processes that lead to the development and decay of RWPs and their representation in Earth system models is crucial for understanding and predicting future extremes.




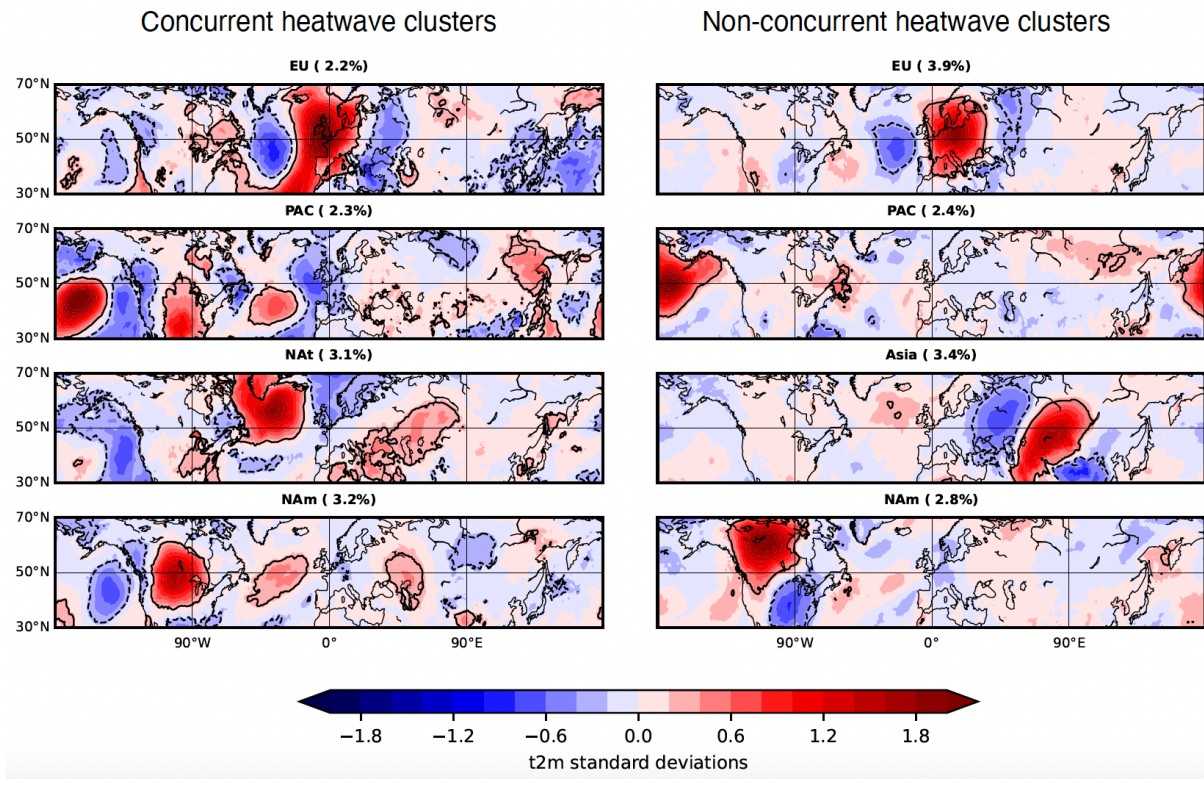

**Figure 1.** Composites of standardized temperature anomalies for the summers of 1959-2021, for the selected concurrent (left column) and non-concurrent (right column) heatwave clusters of the NH. The contour lines +0.3 and -0.3 are given by the black solid and dashed lines, respectively. The regions enclosed by the +0.3 contour indicate the regions where heatwaves occur, corresponding to the regions where the cluster centroids indicate heatwave occurrence.



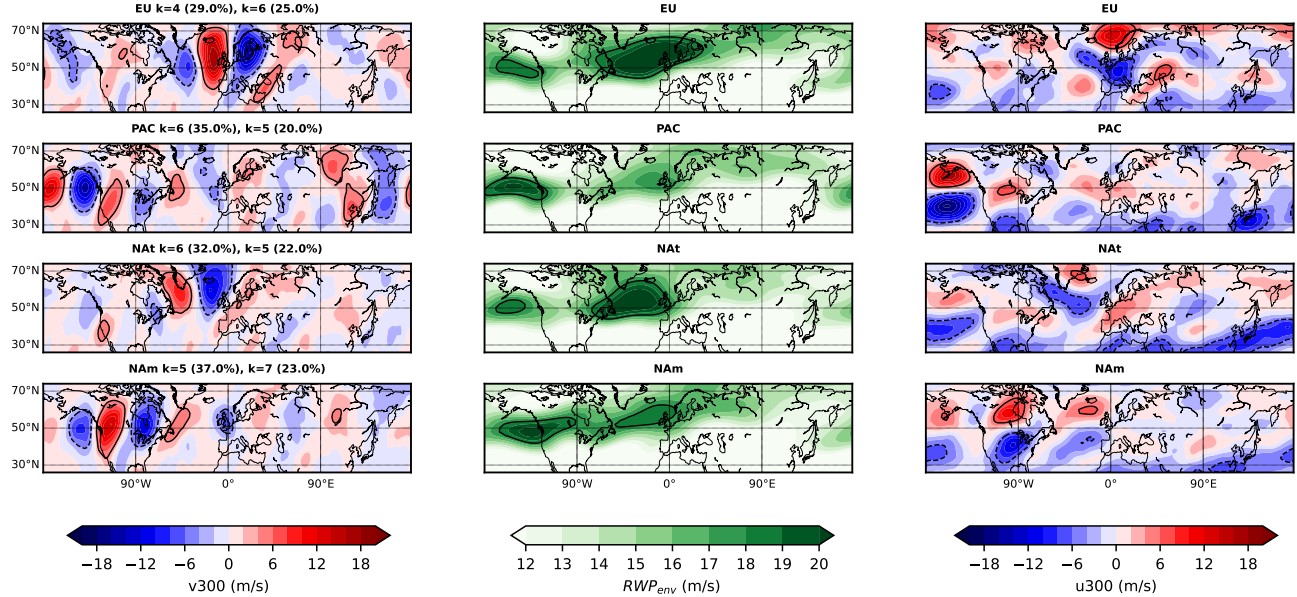

**Figure 2.** Composites of anomalies for v300 (left column), $RWP_{env}$ (middle column), and u300 (right column) for the selected concurrent heatwave clusters of the NH. The clusters shown here correspond to the concurrent clusters shown in Figure 1. The contour levels of $\pm 3$ m/s, +18 m/s, and $\pm 7$ m/s are given for the composites of v300, $RWP_{env}$, and u300, respectively. Positive values in v300 denote stronger southerlies and in u300 denote stronger westerlies than climatology. The two dominant zonal wavenumbers and their frequency are indicated for each heatwave cluster.



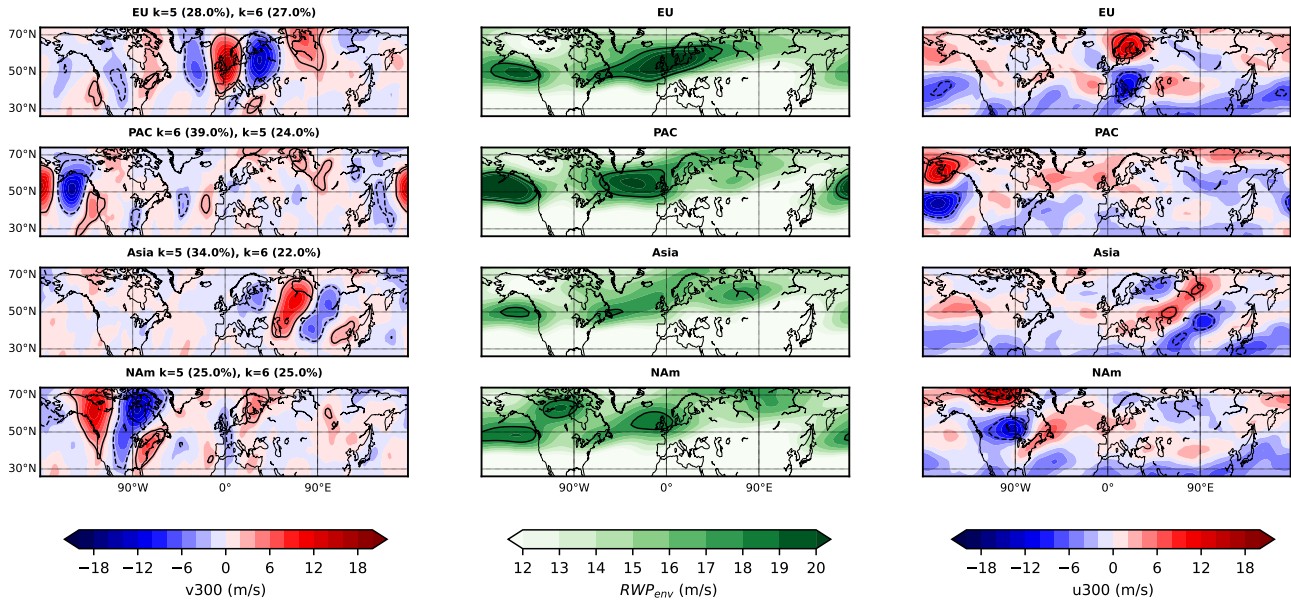

**Figure 3.** Same as Figure 2, but for the selected non-concurrent heatwave clusters of the NH. The clusters shown here correspond to the non-concurrent clusters shown in Figure 1.



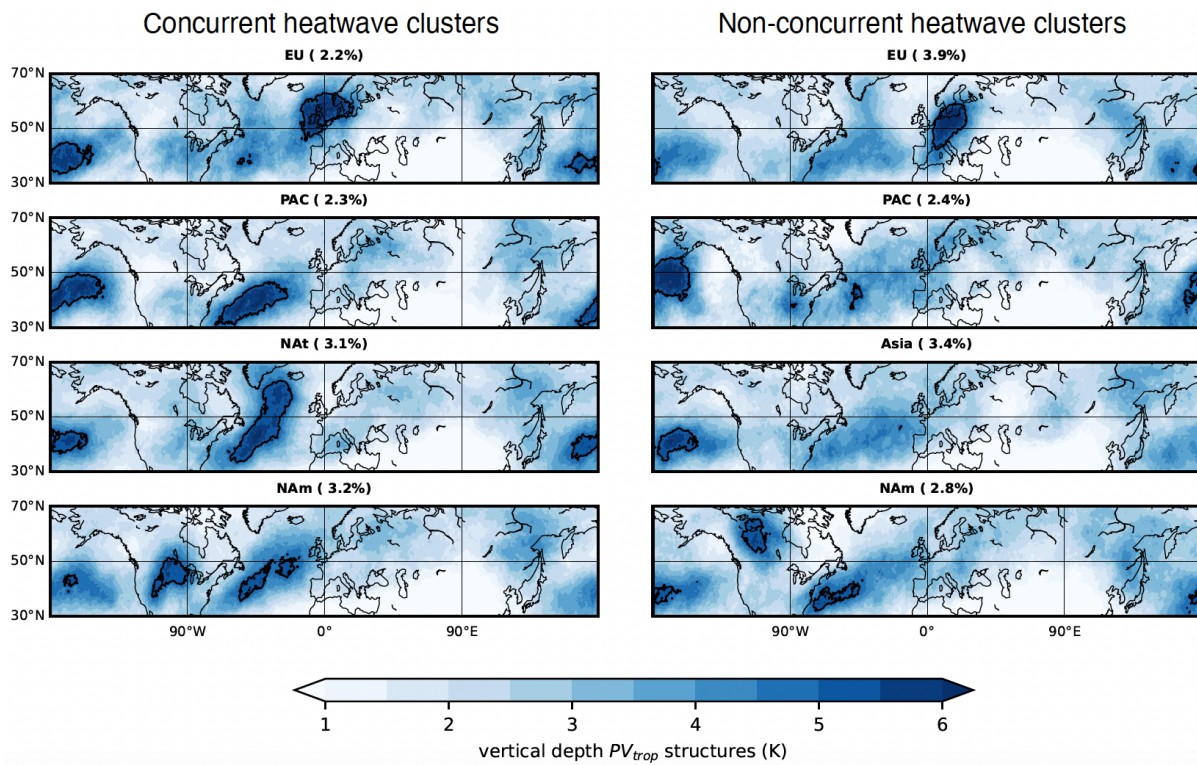

**Figure 4.** Composites of $PV_{trop}$ structures vertical depth (K) for the selected concurrent (left column) and non-concurrent (right column) heatwave clusters of the NH mid-latitudes. The contour value (solid black line) is equal to 5K.





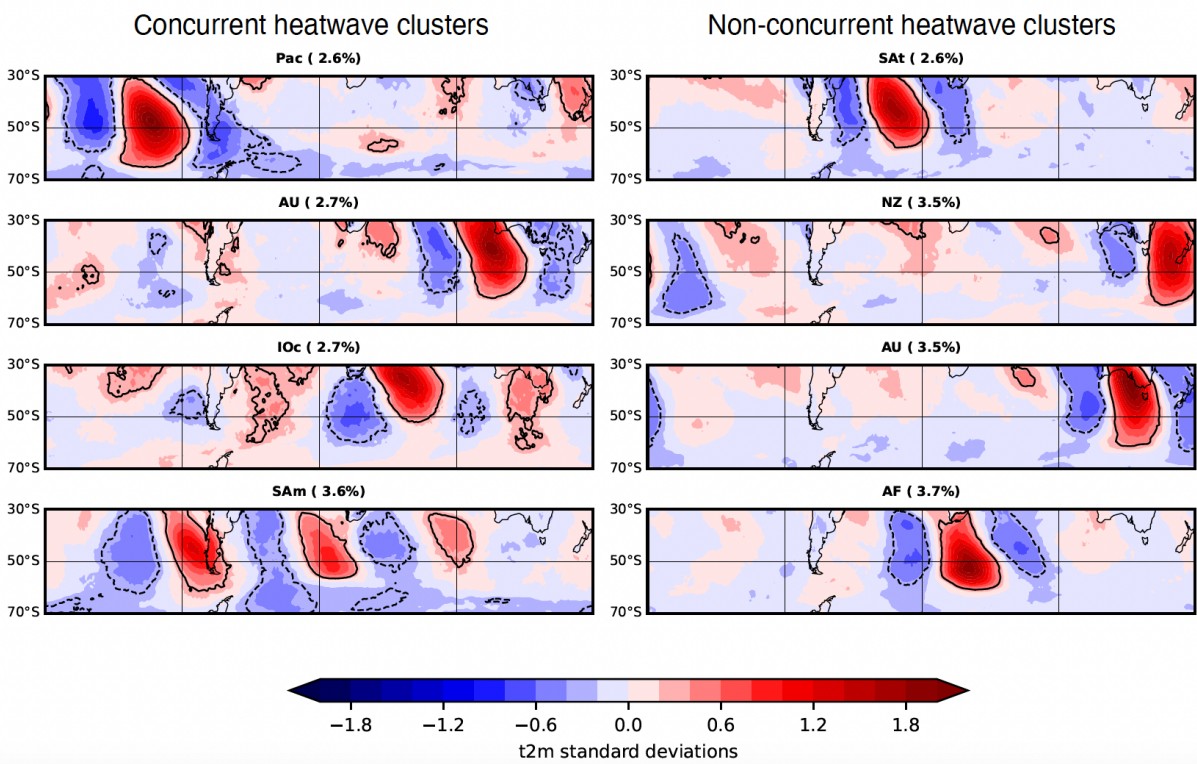

**Figure 5.** Same as Figure 1, but for the SH.



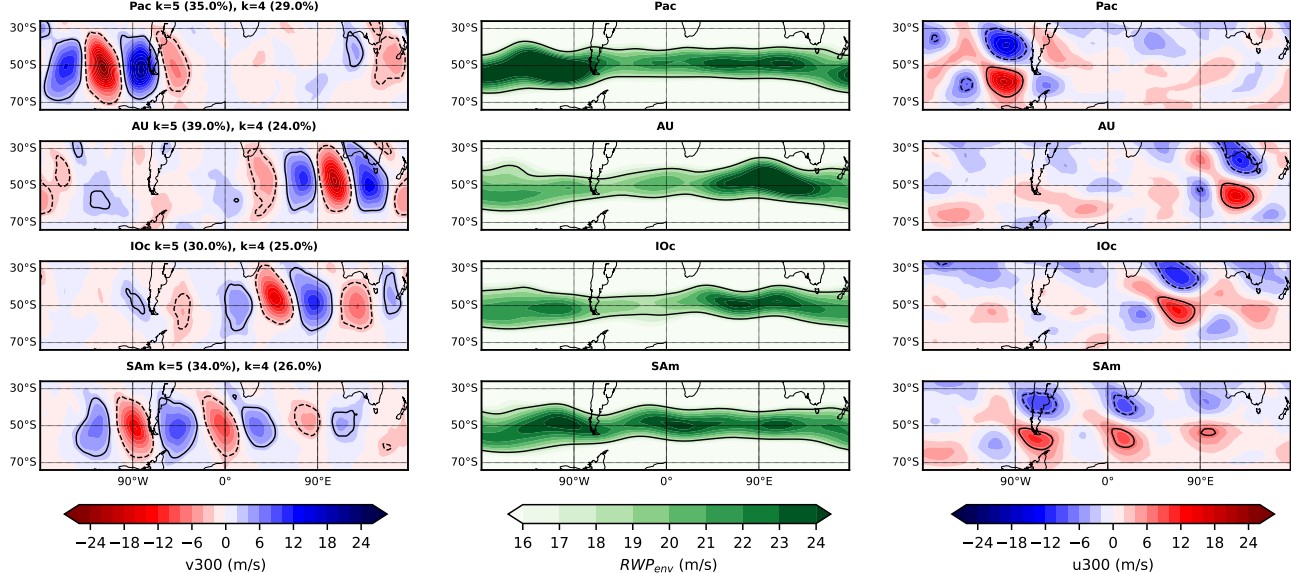

**Figure 6.** Composites of anomalies for v300 (left column), $RWP_{env}$ (middle column), and u300 (right column) for the concurrent heatwave clusters of the SH. The clusters shown here correspond to the concurrent clusters shown in Figure 5. The contour levels of $\pm 3$ m/s, $+18$ m/s, and $\pm 7$ m/s are given for the composites of v300, $RWP_{env}$, and u300, respectively. Positive values in v300 denote stronger northerlies and in u300 denote stronger westerlies than climatology. The two dominant zonal wavenumbers, being the ones with the highest amplitude, and their frequency are indicated for each heatwave cluster.





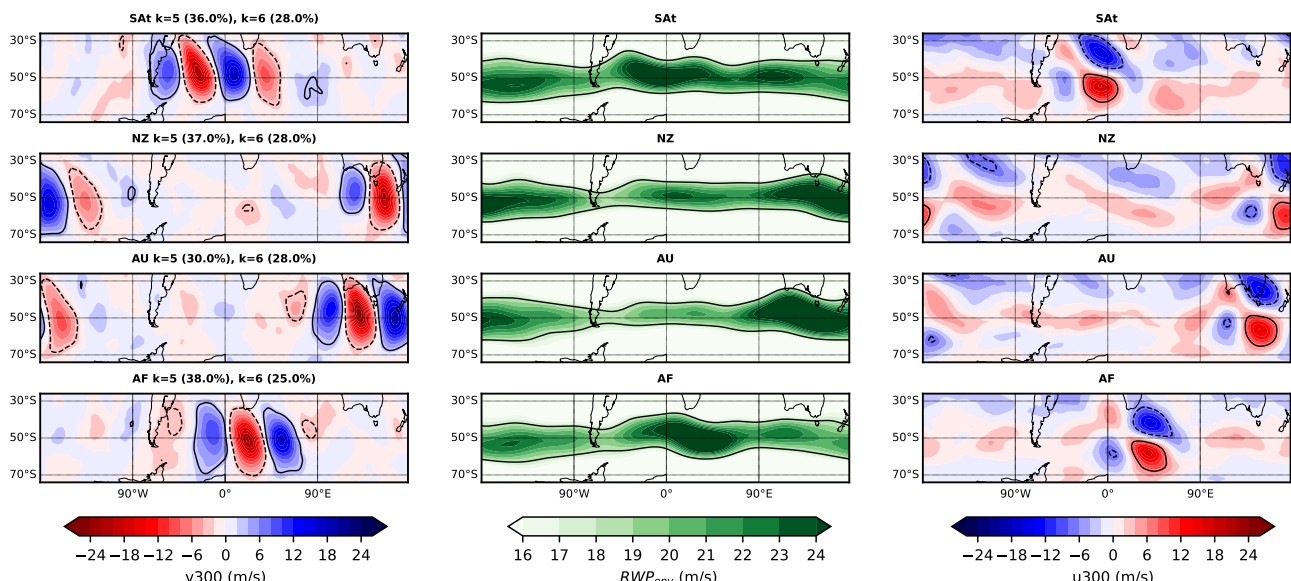

**Figure 7.** Same as Figure 6, but for the selected non-concurrent heatwave clusters of the SH. The clusters shown here correspond to the non-concurrent clusters shown in Figure 5.



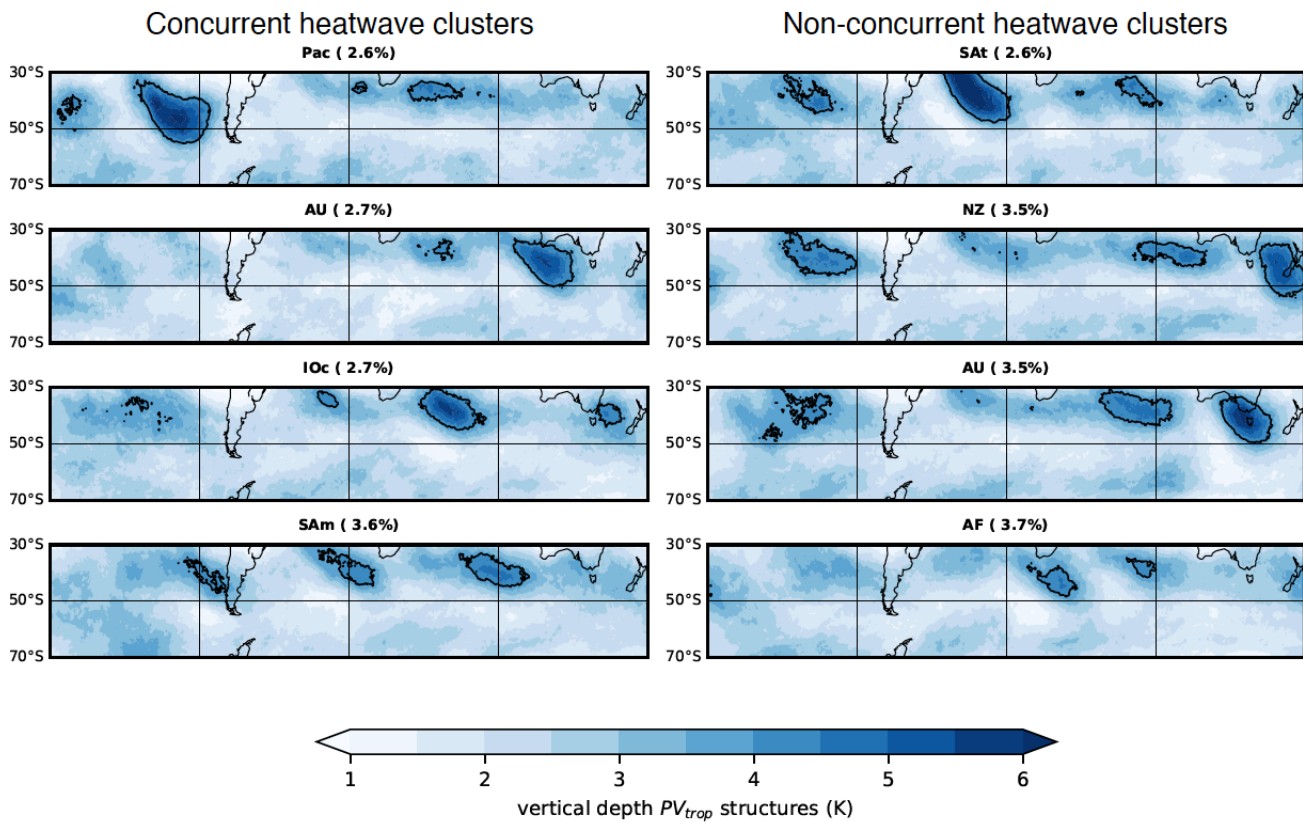

**Figure 8.** Composites of $PV_{trop}$ structures vertical depth (K) for the selected concurrent (left column) and non-concurrent (right column) heatwave clusters of the SH mid-latitudes. The contour value is equal to 4K.



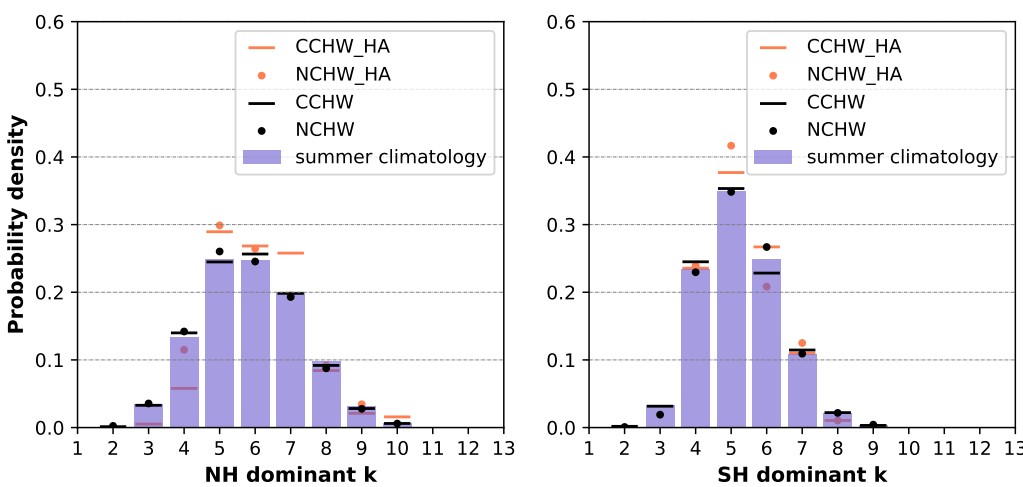

**Figure 9.** Probability density of dominant zonal wavenumbers (i.e., wavenumbers with highest amplitude) for the NH (left) and the SH (right) mid-latitudes. The climatological summer probability is given by the purple bar. The probability during concurrent (CCHW) and non-concurrent heatwaves (NCHW) is given by the black line and black circle, respectively. The probability during concurrent and non-concurrent heatwaves with high wave amplitude requirement, i.e., CCHW_HA and NCHW_HA, is given by the orange line and orange circle, respectively.



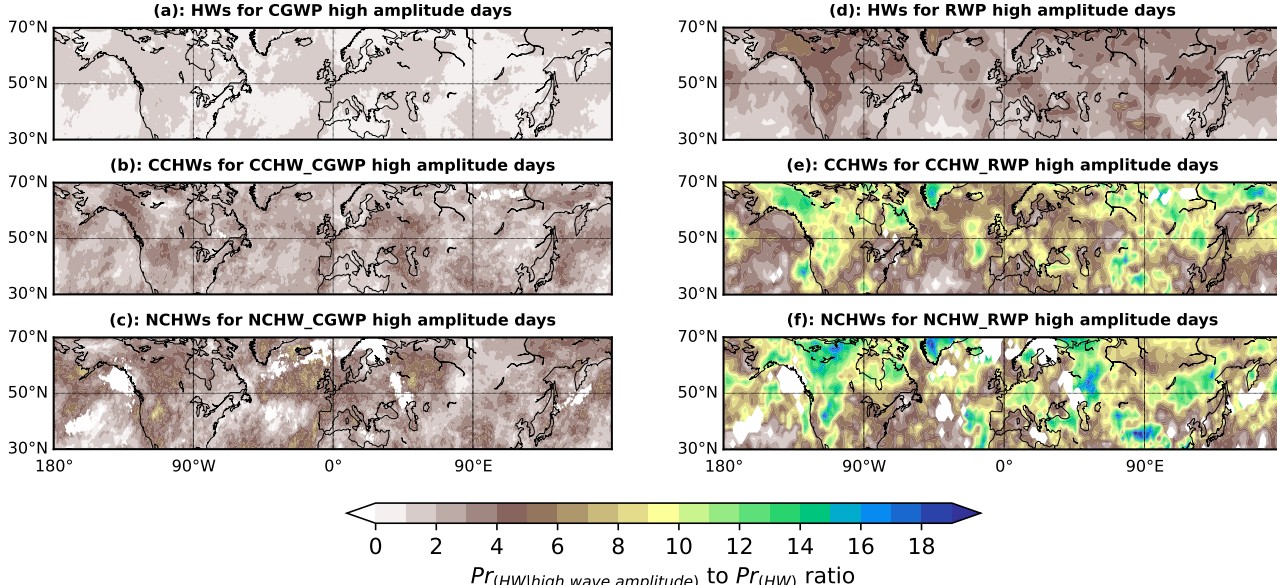

**Figure 10.** Ratio of heatwave conditional probabilities during high global (CGWP; left column) and high local ($RWP_{env}$; right column) wave amplitude to heatwave probabilities. The rows represent from top to bottom the probability ratios computed for all heatwave (HW) days with respect to all high amplitude days, for concurrent heatwave (CCHW) days with respect to high amplitude days of concurrent heatwave clusters, and for non-concurrent heatwave (NCHW) days with respect to high amplitude days of non-concurrent heatwave clusters. The regions masked with probabilities equal to zero represent regions where the frequency of the respective concurrent or non-concurrent heatwave days is less than 1 percent of the total sample size. The same sample size criterion was used to determine the heatwave clusters investigated in the current study.



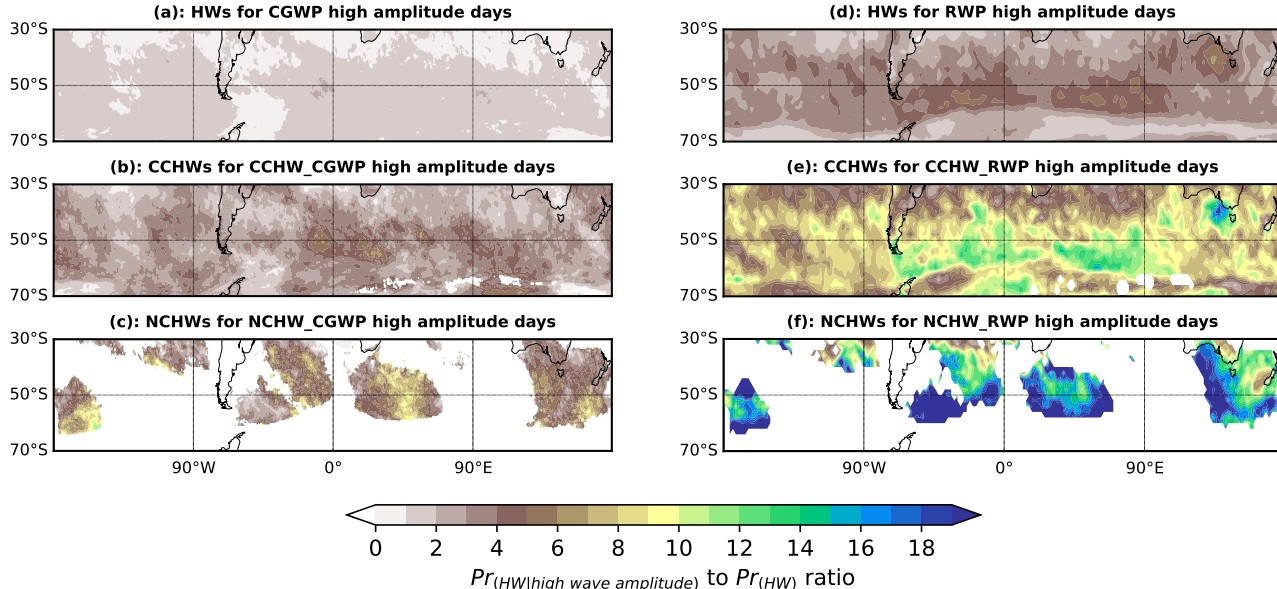

**Figure 11.** Same as Figure 10, but for the SH.



**Figure A1.** Trends of v300 (m/s per summer; a-d) and t2m (K per summer; e-h) for the NH (a, b, e, f) and the SH (c, d, g, h) for the pre-satellite (1959-1979; left column) and the satellite era (1980-2021; right column).

## Appendix A: RWPs, CGWP, and heatwaves in the Northern and Southern Hemisphere mid-latitudes

The analysis was repeated for 1. linearly detrending the full period, 2. detrending step-wise in 3 periods (1959-1979, 1980-2000, 2001-2021), and 3. removing a 5-year running mean. The results reported are similar for all the methods and for both hemispheres.





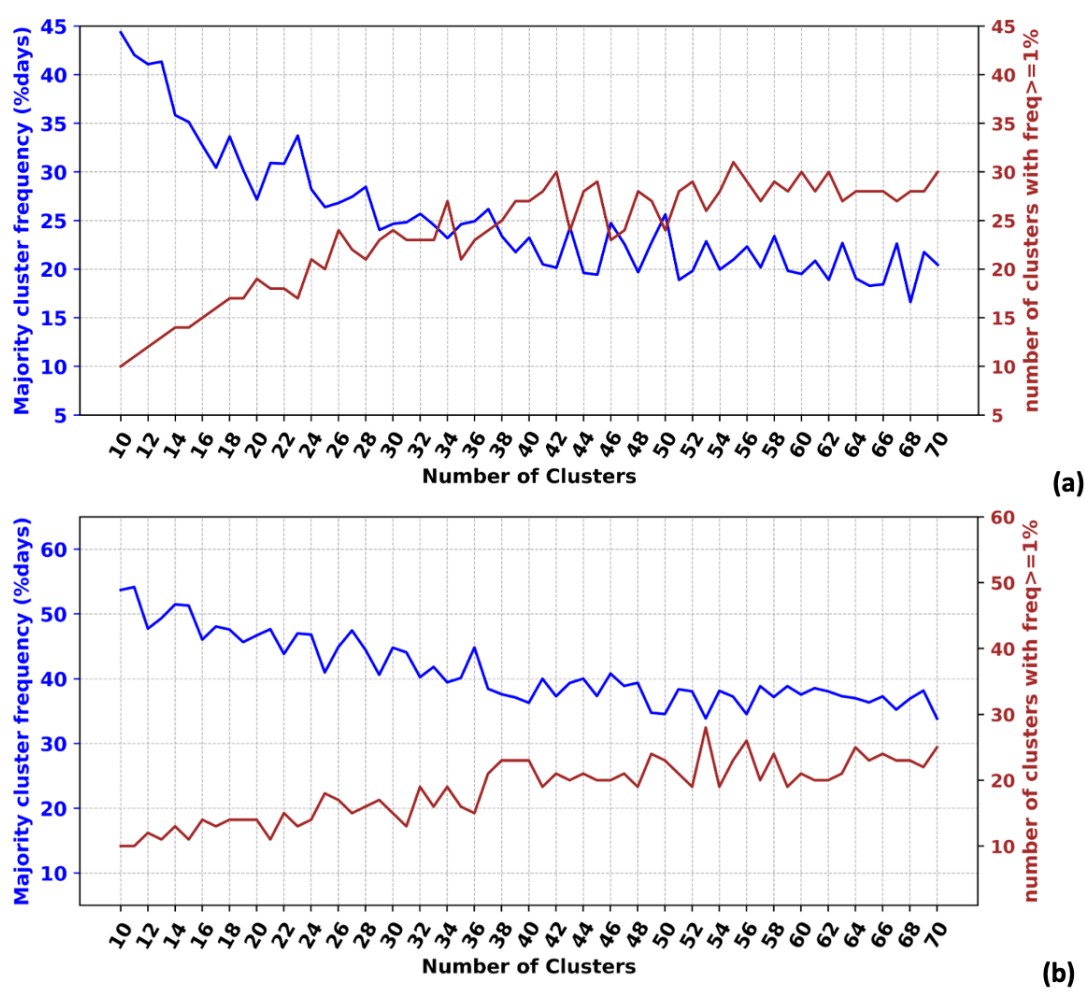

**Figure A2.** Number of clusters with frequency of occurrence more than or equal to 1% (brown line) and frequency of occurrence of the majority cluster (blue line) for the (a) NH and (b) SH.





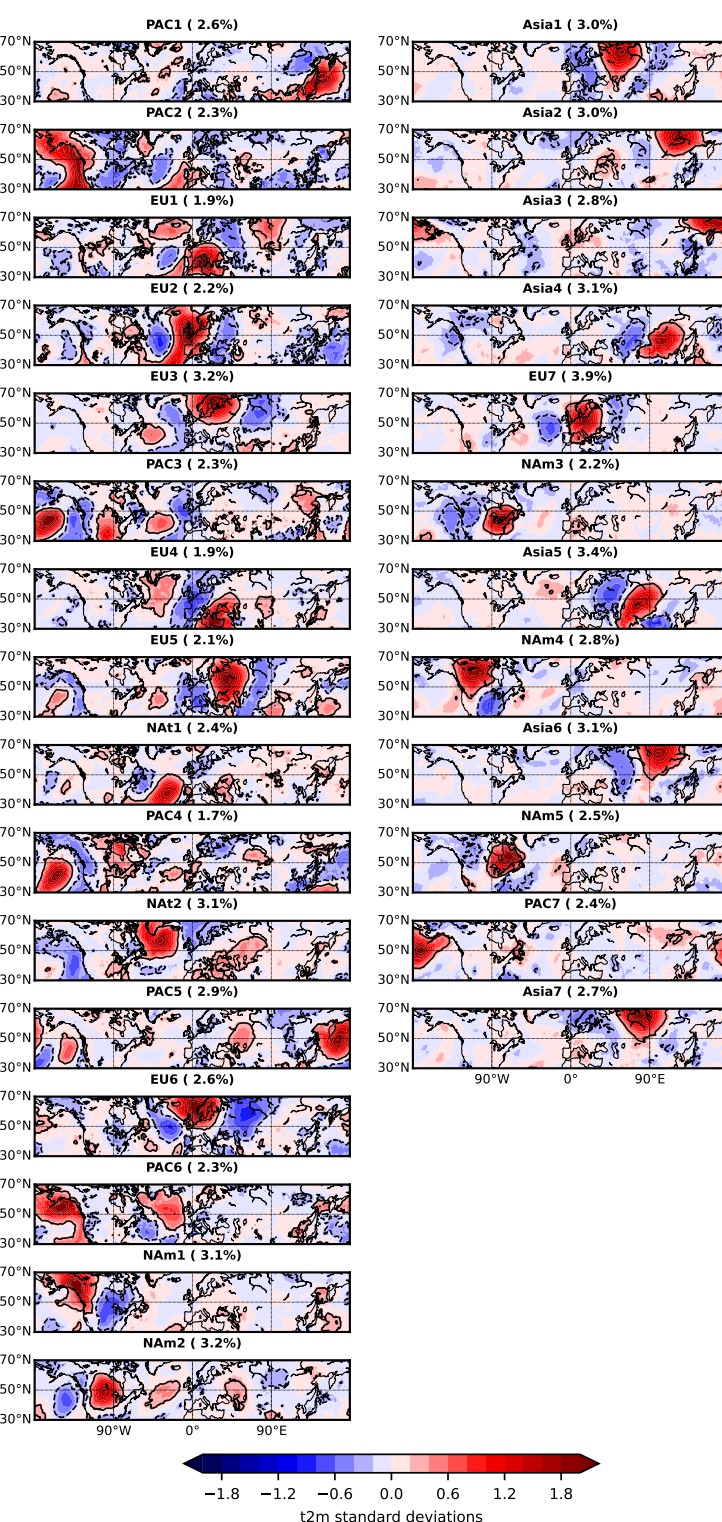

**Figure A3.** Composites of standardized temperature anomalies for the summers of 1959-2021, for all the concurrent (left column) and non-concurrent (right column) heatwave clusters identified in the NH. The contour lines +0.3 and -0.3 are given by the black solid and dashed lines, respectively. The regions enclosed by the +0.3 contour indicate the regions where heatwaves occur and correspond to the regions where the cluster centroids indicate heatwave occurrence.



**Figure A4.** Composites of anomalies for v300 (left column), $RWP_{env}$ (middle column), and u300 (right column),



**Figure A4.** for all the concurrent heatwave clusters identified in the NH. The contour levels of ±2 m/s, +16 m/s, and ±4 m/s are given for the composites of v300, $RWP_{env}$, and u300, respectively. Positive values in v300 denote stronger southerlies and in u300 denote stronger westerlies than climatology. The two dominant zonal wavenumbers and their frequency are indicated for each heatwave cluster.

**Figure A5.** Same as Figure A4, but for all the non-concurrent heatwave clusters identified in the NH.





**Figure A6.** Composites of anomalies for v300 (left column), $RWP_{env}$ (middle column), and u300 (right column),



**Figure A6.** for all the concurrent heatwave clusters identified in the NH after applying the criterion of **3-day heatwave persistence**. The contour levels of $\pm 2$ m/s, +16 m/s, and $\pm 4$ m/s are given for the composites of v300, $RWP_{env}$, and u300, respectively. Positive values in v300 denote stronger southerlies and in u300 denote stronger westerlies than climatology. The two dominant zonal wavenumbers and their frequency are indicated for each heatwave cluster.

**Figure A7.** Same as Figure A6, but for the non-concurrent heatwave clusters identified in the NH after applying the criterion of **3-day heatwave persistence**.

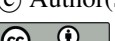



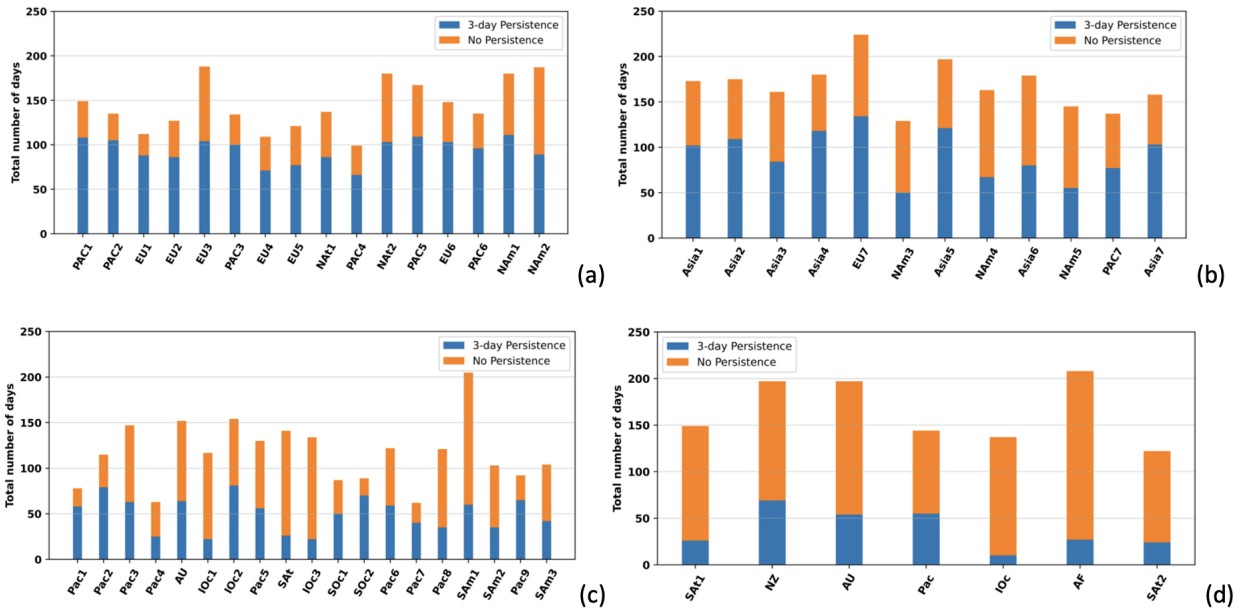

**Figure A8.** Total number of heatwave days for concurrent clusters (a, c) and non-concurrent clusters (b, d) for the NH (a, b) and SH (c, d). The number of days is given by the top of the bar for heatwaves having at least 3-day persistence (blue bar), 2-day persistence (orange bar), and no persistence (green bar).



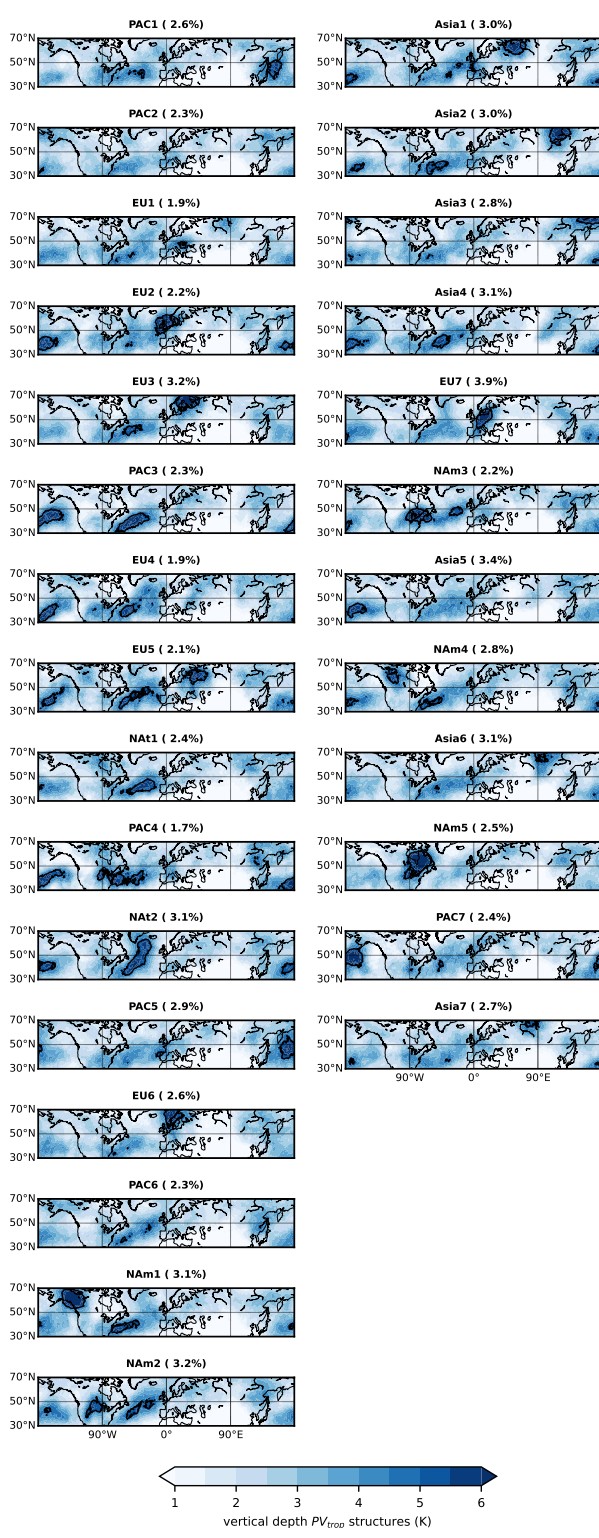

**Figure A9.** Composites of $PV_{trop}$ structures vertical depth (K) for the selected concurrent (left column) and non-concurrent (right column) heatwave clusters of the NH. The contour value is equal to 5K.



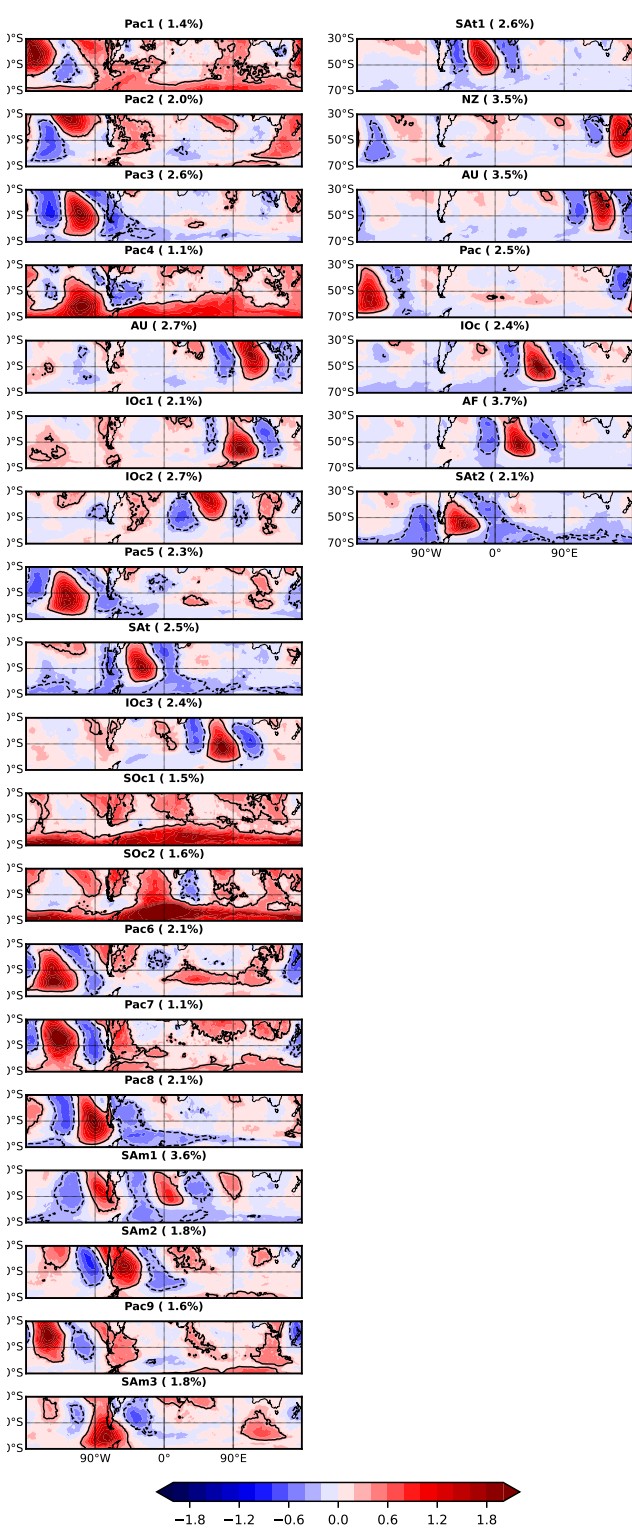

**Figure A10.** Composites of standardized temperature anomalies for all the concurrent (left column) and non-concurrent (right column) heatwave clusters identified in the SH. The regions enclosed by the +0.3 contour indicate the regions where heatwaves occur and correspond to the regions where the cluster centroids indicate heatwave occurrence.





**Figure A11.** Composites of anomalies for v300 (left column), $RWP_{env}$ (middle column), and u300 (right column) for all the concurrent



**Figure A11.** heatwave clusters identified in the SH. The contour levels of ±4 m/s, +18 m/s, and ±6 m/s are given for the composites of v300, $RWP_{env}$, and u300, respectively. Positive values in v300 denote stronger northerlies and in u300 denote stronger westerlies than climatology. The dominant zonal wavenumbers and their frequency are indicated for each heatwave cluster.

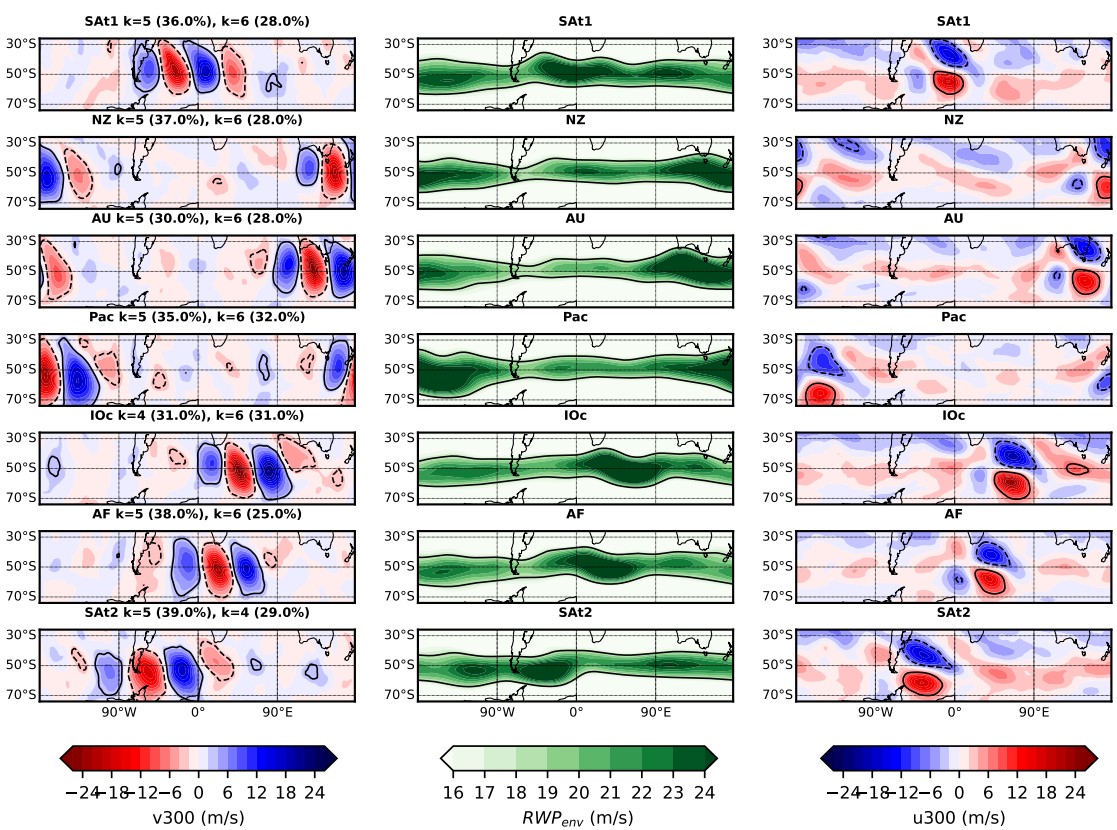

**Figure A12.** Same as Figure A11, but for all the non-concurrent heatwave clusters identified in the SH.





**Figure A13.** Composites of anomalies for v300 (left column), $RWP_{env}$ (middle column), and u300 (right column),



**Figure A13.** for all the concurrent heatwave clusters identified in the SH after applying the criterion of **2-day heatwave persistence**. The contour levels of $\pm 4$ m/s, $+18$ m/s, and $\pm 6$ m/s are given for the composites of v300, $RWP_{env}$, and u300, respectively. Positive values in v300 denote stronger northerlies and in u300 denote stronger westerlies than climatology. The two dominant zonal wavenumbers and their frequency are indicated for each heatwave cluster.

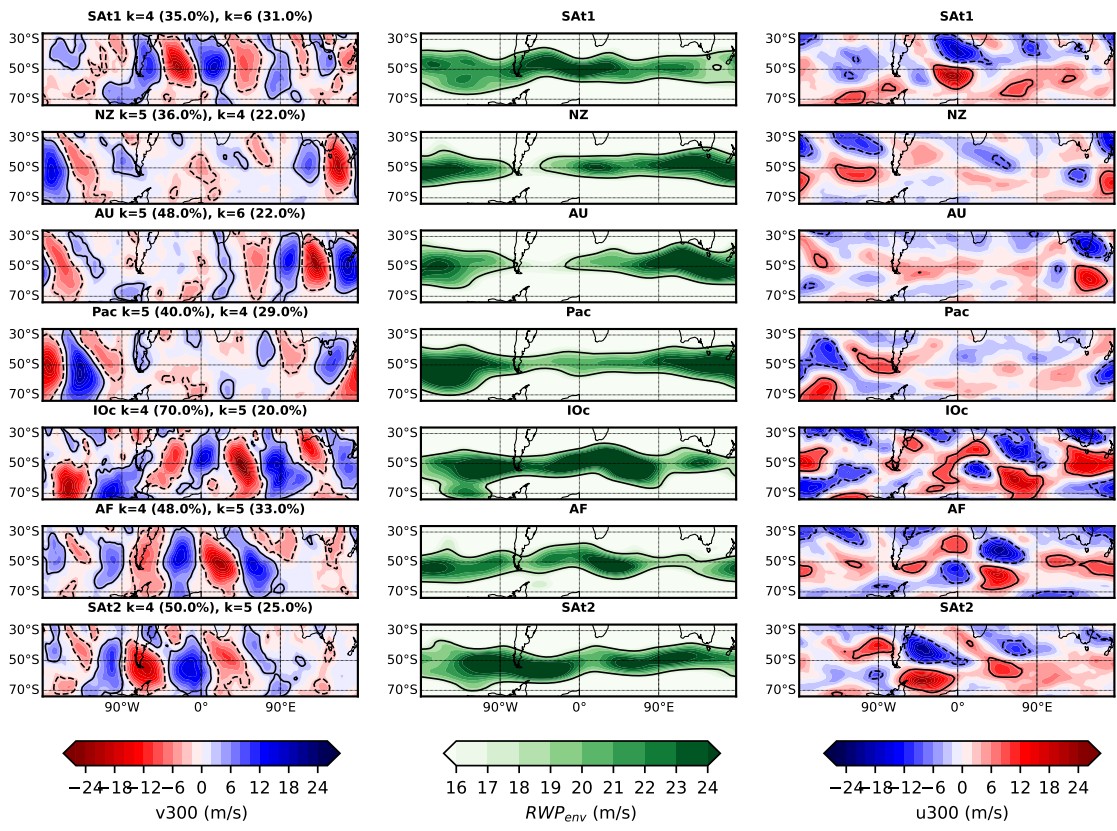

**Figure A14.** Same as Figure A13, but for all the non-concurrent clusters identified in the SH after applying the criterion of **3-day persistence**.



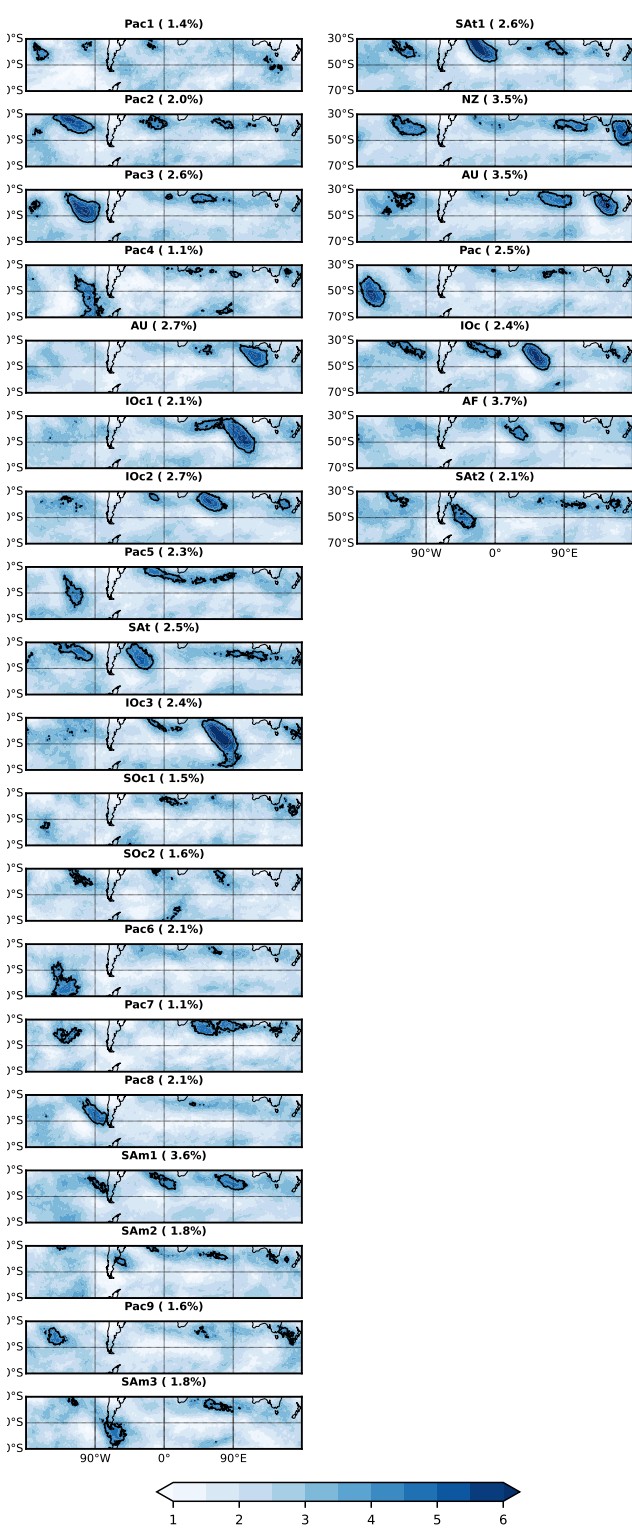

**Figure A15.** Composites of $PV_{trop}$ structures vertical depth (K) for the selected concurrent (left column) and non-concurrent (right column) heatwave clusters of the SH. The contour value is equal to 4K.



*Code and data availability.* The code for reproducing the figures of this manuscript is available in GitHub at: https://github.com/marpyr/ 400 Rossby-wave-packets-driving-concurrent-and-non-concurrent-heatwaves-in-the-NH-and-SH.

*Author contributions.* Maria Pyrina: conceptualization; data curation; formal analysis; analytic calculations; investigation; methodology; visualization; writing – original draft; writing – review and editing

Wolfgang Wicker: methodology; writing – review and editing

Andries Jan de Vries: data acquisition; methodology; writing – review and editing

Georgios Fragkoulidis: methodology; writing – review and editing

Daniela I.V. Domeisen: conceptualization; funding acquisition; supervision; writing – review and editing.

*Competing interests.* At least one of the (co-)authors is a member of the editorial board of Weather and Climate Dynamics.

*Acknowledgements.* The authors would like to thank the Copernicus Climate Change Service (C3S) for making available the ERA5 data 410 (Hersbach et al., 2020). Support from the Swiss National Science Foundation through project PP00P2_198896 to M.P. and D.D. is gratefully acknowledged. This project has received funding from the European Research Council (ERC) under the European Union's Horizon 2020 research and innovation programme (grant agreement No. 847456). M.P. acknowledges financial support from the Collaborative Research on Science and Society (CROSS) Program of EPFL and UNIL.



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
