# Peer review of "Rossby wave packets driving concurrent and non-concurrent heatwaves in the Northern and Southern Hemisphere mid-latitudes"

_EGUsphere, 2023_

## Editor Comment (EC1)

Recommendation Pyrina_etal_egusphere-2023-3088

I have now received two solicited anonymous reviews and one unsolicited review of Pyrina et al (WCD 2023-3088). Two out of three reviewers state the paper is not suitable for publication and the third reviewer also has major concerns. As such, I am rejecting the paper in hopes that you will be able to use the reviewers' comments to craft a new manuscript that addresses their major concerns.

All three reviews are concerned that there isn't a sufficient description of the methodology. In particular, (i) the reviewers express concern that identification of heatwaves by cluster analysis is likely to be sensitive to choices in way the clustering is performed and that the paper must demonstrate how these choices affect the results, and (ii) the reviewers did not see a compelling argument for the parameter choices that were used to classify heatwaves into concurrent and non-concurrent events (e.g., how sensitive are the results to the threshold value (> 0.3 sigma) that is used to define a heatwave?).

As you emphasize in your manuscript, the structure of the composite circulation and RWP envelop ($RWP_{env}$) for the concurrent and non-concurrent heatwaves are very similar – and heatwaves in both cases appear to be associated Rossby wave packets and not circumglobal Rossby waves (CGRWs). However, two of the reviewers are skeptical of the metrics used to identify the CGRWs and their relationship to the heatwaves identified using the cluster analysis, and they point out a literature that demonstrates a strong relationship between CGRWs and concurrent heat waves. To draw any firm conclusions concerning the relative importance of CGRWs and isolated Rossby wave packets for concurrent heat waves, a thorough analysis should be presented that demonstrates the schemes to distinguish CGRW from $RWP_{env}$ are not sensitive to the choice of wavenumbers used or even to the choice of index (e.g., one could make an argument that the CGRW are best identified by the r.m.s. of the squared amplitude of wavenumbers k 4-15 rather than the sum of the amplitudes).